# Lubrication and Sensory Properties of Emulsion Systems and Effects of Droplet Size Distribution

**DOI:** 10.3390/foods10123024

**Published:** 2021-12-06

**Authors:** Qi Wang, Yang Zhu, Zhichao Ji, Jianshe Chen

**Affiliations:** Laboratory of Food Oral Processing, School of Food Science and Biotechnology, Zhejiang Gongshang University, Hangzhou 310018, China; qiwong1995@163.com (Q.W.); jjzzcc1998@163.com (Z.J.); jschen@zjgsu.edu.cn (J.C.)

**Keywords:** oral lubrication, oral tribology, food emulsions, droplet size distribution, food sensory

## Abstract

The functional and sensory properties of food emulsion are thought to be complicated and influenced by many factors, such as the emulsifier, oil/fat mass fraction, and size of oil/fat droplets. In addition, the perceived texture of food emulsion during oral processing is mainly dominated by its rheological and tribological responses. This study investigated the effect of droplet size distribution as well as the content of oil droplets on the lubrication and sensory properties of o/w emulsion systems. Friction curves for reconstituted milk samples (composition: skimmed milk and milk cream) and Casein sodium salt (hereinafter referred to as CSS) stabilized model emulsions (olive oil as oil phase) were obtained using a soft texture analyzer tribometer with a three ball-on-disc setup combined with a soft surfaces (PDMS) tribology system. Sensory discrimination was conducted by 22 participants using an intensity scoring method. Stribeck curve analyses showed that, for reconstituted milk samples with similar rheological properties, increasing the volume fraction of oil/fat droplets in the size range of 1–10 µm will significantly enhance lubrication, while for CSS-stabilized emulsions, the size effect of oil/fat droplets reduced to around 1 µm. Surprisingly, once the size of oil/fat droplets of both systems reached nano size (d_90_ = 0.3 µm), increasing the oil/fat content gave no further enhancement, and the friction coefficient showed no significant difference (*p* > 0.05). Results from sensory analysis show that consumers are capable of discriminating emulsions, which vary in oil/fat droplet size and in oil/fat content (*p* < 0.01). However, it appeared that the discrimination capability of the panelist was significantly reduced for emulsions containing nano-sized droplets.

## 1. Introduction

Awareness of the critical importance of diet on human health and wellness is growing among consumers, especially due to well-reported evidence showing excessive calorie intake could lead to occurrence of obesity, high blood pressure, and cardiovascular diseases [1,2]. The food industry has, therefore, been actively seeking new techniques and food formulations of calorie reduction, such as low-fat or fat-free versions of traditional food products that maintain their sensory properties [3,4,5]. However, removing or reducing the amount of oil/fat in food systems is not as easy as it seems to be, simply due to the multiple roles of oil/fat in food formulations, particularly when the oil/fat is in a dispersed status and functions as a structural component of the matrix. Any small change in the mass fraction and droplet size distribution of oil droplets, the type and concentration of emulsifier, will alter the physico-chemical properties of the emulsion [3]. For example, fat droplets can affect the appearance (optical properties) [6], flavor characteristics (molecular distribution) [7,8], texture properties (rheology, tribology) [9], and shelf life (stability) [10] of food emulsions. In relation to texture and mouthfeel, oil/fat provides at least three basic functional roles: (1) as a filler ingredient to alter microstructure and rheology of such systems; (2) adsorption on the tongue surface for enhanced oil/fat sensation; and (3) as a particulate component to improve lubricating properties [11,12].

In past decades, rheology has been taken granted as a ‘gold standard’ instrumental technique for mapping or predicting the perceived texture and mouthfeel of liquid products [13,14,15]. However, as liquid and semisolid food are squeezed and rubbed between the tongue and hard plate during oral processing, the traditional rheology approach was found limited in explaining the perception of such as smoothness, creaminess, greasiness, etc. Instead, growing evidence shows that the lubrication behavior in the oral cavity plays a dominating role in influencing oral sensation and sensory perception [16,17,18]. In a typical tribological measurement, consisting of two surfaces lubricated by a Newtonian fluid, a Stribeck curve can be obtained to give the relationship between the friction coefficient and the thickness of the lubricating film [19,20,21]. If one of the surfaces is soft and deformable (i.e., tongue against hard palate), it is a soft-contact lubrication. Three different lubrication regimes have been established for a typical Stribeck curve: the boundary regime at low thickness of the lubricating film, the mixed regime at medium thickness, and the hydrodynamic regime at high thickness. The underlying mechanisms of lubrication regimes have been well described by a number of researchers [17,22,23,24].

Since the sensory mouthfeel of a food emulsion is closely linked to its rheological and tribological properties, it is now commonly accepted that oil/fat volume fraction, size of oil/fat droplets, and the viscosity of the emulsion will have direct impacts on the lubrication behavior and the perception of fat-related sensory attributes of such systems [25]. Malone et al. confirmed many years ago that correlations existed between sensory-perceived slipperiness and friction coefficients for biopolymer solutions [26]. A recent study by authors’ group investigated the lubrication properties of emulsions stabilized by different emulsifiers [27], observing a good correlation between the sensation of finger and oral smoothness and the measured friction coefficient, which suggests that smoothness perception is mainly driven by tactile sensation. Chojnicka-Paszun et al. demonstrated a significant correlation between creamy attributes and the friction coefficient measured on soft rubber surfaces [28]. They suggested that creaminess is best predicted by the friction obtained at low speeds (comparable with the speed in the mouth) that corresponded to the boundary lubrication regime. However, the literature research findings are not always straightforward and consistent. For example, some researchers noted that the oil/fat volume fractions seem to have little effect on the lubrication behavior in the boundary regime, while increasing the oil/fat volume fractions of o/w emulsion system, resulting in an earlier shift from boundary to mixed regime [10,27]. A study by Laguna et al. showed that in vitro lubrication tests failed to distinguish milk samples containing different mass fractions of fat while untrained panelists were able to [25]. In addition to the oil/fat mass fraction, it has been found that the coalescence of fat droplets during oral processing may led to a decrease in friction coefficient and an increased perception of fat-related sensory attributes, such as creaminess [29]. An increased, effective oil/fat volume fraction and the increased viscosity could be the reasons behind this observation [7]. The presence of saliva may increase the friction of the system through the flocculation phenomenon for positively charged droplets or promote the adherence of saliva proteins to the substrate for negatively charged emulsions, both resulting in a stronger correlation of perceived graininess and fattiness to tribological properties [30,31,32].

Most of the previous tribological studies examined the effect of the oil/fat volume fraction on the lubrication behavior of liquid dairy products or model emulsion systems. However, the size effect of dispersed fat droplets on the lubrication behavior and sensory perception appears to be somewhat contradictory in the literature. It has been speculated that decreasing the oil droplet size at constant oil volume fraction could increase the perceived creaminess [33], while others discovered contradictory phenomena, showing that the clustered emulsion has significantly higher creaminess intensity than single droplets [32]. Therefore, in this study, we choose two o/w emulsion systems as major research objects (commercial dairy products composed of skimmed milk and milk cream; model emulsion composed of CSS and olive oil), with the objective to reveal the effect of the size of the fat droplet on the in vitro lubrication behavior of emulsions and then to investigate possible applications of the tribological approach in assessing the texture perception of food emulsions.

## 2. Materials and Methods

### 2.1. Sample Preparation

Pasteurized skimmed milk with fat content of 0% (*w*/*w*) (Avonmore, China Resources Wufeng Distribution (Shenzhen) Co., Ltd., Shenzhen, China) and milk cream with fat content of 35% (President, Angliss (Shanghai) Food Co., Ltd., Shanghai, China) were mixed in order to obtain reconstituted milk samples with specified fat contents of 0.3, 0.7, 2.0, 3.5, and 7.5 wt.%, respectively. The reconstituted milk samples (o/w emulsion) were first sheared using an Ultra-Turrax (Polytron, Kinematica AG, Lucerne, Switzerland) at 3000 rpm for 120 s (first stage) and homogenized using a high-pressure homogenizer (HPH—second stage). Samples with different fat droplet size distribution were obtained by adjusting the operating pressures (bar) and time (s) of the high-pressure homogenizer (AH-BASIC, ATS, Canada). The thickened samples were obtained by adding 1.4 wt.% commercial thickener (Ourdiet Swallow^®^, Guangzhou Ourdiet Biotechnology Co., Ltd., Guangzhou, China) into the prepared reconstituted milk samples at room temperature under magnetic stirring, then treated by Shear homogenizer at 3000 rpm for 120 s, accelerating the dissolution of thickener. In accordance with the specification of the manufacturer, the main ingredients of the thickener are xanthan gum (60%) and maltodextrin (28%).

The abbreviation used in the following text: reconstituted milk samples with different fat content homogenized under an Ultra-Turrax were labeled as ‘Fat content-Shear’, but samples which were homogenized under high pressure were labeled as ‘Fat content-HPH’. Samples with the addition of thicker were labeled as ‘Fat content-thicken-Shear/HPH’.

Model oil-in-water emulsions, consisting of 7.5 wt.% olive oil and 92.5 wt.% aqueous phase containing 1 wt.% Casein sodium salt (Sigma Aldrich, Auckland, New Zealand), were prepared by pre-homogenizing the ingredients using an Ultra-Turrax. Emulsions of different fat droplet size were obtained by adjusting the operating pressures (bar) and time (s) of the high-pressure homogenizer.

### 2.2. Droplet Size Studies

The droplet size distribution of all the samples was measured using a laser diffraction particle size analyzer (Mastersizer 3000, Malvern Instruments, Ltd., Worcestershire, UK). Freshly prepared emulsion samples were gently stirred and diluted by adding small droplets (about 30 µL) into a measurement chamber containing water, until the instrument gave an optimum obscuration rate around 10%. The optical model used to resolve the droplet size distribution from scattering data used a refractive index of 1.33 for the aqueous phase and 1.43 for the fat/oil phase (emulsions). Analysis was set for a regular round droplet and measured in quintuplicate for determinations for all samples. An averaged size distribution was obtained for each sample, and both the Sauter mean d_3,2_ and the 90th percentile (d_90_) were used to compare differences in the surface-weighted droplet size and distribution of coarser droplet, respectively.

### 2.3. Flow Behavior and Apparent Viscosity

The rheological properties of reconstituted milk and model oil-in-water emulsions were measured using a rotational rheometer (Discovery Hybrid Rheometer-2, TA Instrument, New Castle, DE, USA). All the measurments were carried out by using a cone-and-plate geometry (diameter: 40 mm, angle: 2.017°, operating gap: 55 µm) under the flow ramp, with shear rate ranging between 0.01 and 1000 s^−1^. All the samples were equilibrated at 25 °C before measurement. After being loaded, each sample was equilibrated again for 60 s at 25 °C before test was started. All measurements were performed in triplicate. The results indicated that samples of both types are typical non-Newtonian and, therefore, the apparent viscosity at 50 s^−1^ was used for the analysis of the tribological data.

### 2.4. Tribology Measurement

Tribological measurements were carried out using a Soft TA-Tribometer (STAT). This device was developed based on a commercial texture analyzer TA, XTPlus (Stable Micro Systems, Surrey, UK), with a specially designed fixture attached to it. Details of an experimental setup have been given previously [34], following the operating principles described before [21]. Figure 1 shows the schematic diagram of the used tribometer; briefly, a soft probe containing three PDMS hemispheres was used to represent the hard palate, and PDMS substrate was used to represent the human tongue surface. The temperature of 25 °C was chosen for tribological measurements, to be in line with the temperature at which the samples were prepared, and Stribeck curves were obtained using a set of sliding speed between 0.1 and 20 mm/s. A normal force of 0.0394 N was applied for tribology measurement. For each measurement, about 6 mL of test sample was transferred onto the surface of a silicone elastomer using a disposable pipet, giving a thickness of fluid film around 2.5 mm. Care must be taken to ensure no air bubble was created during sample transition.

Test data (force, distance, and speed, as well as time) were recorded automatically by the Exponent software (Exponent, Stable Microsystems, version 6.1.9.0, London, UK). Dividing the friction force (N) by the surface load (N) gives the apparent friction coefficient. However, for convenience, all apparent friction coefficients were expressed as friction coefficient in this work, a general practice, which is widely accepted in the literature. Results were the average ± the standard deviation of at least three replicate runs conducted for each experiment.

### 2.5. Sensory Analysis

Reconstituted milk and thickened samples were evaluated by 22 pre-trained panelists (11 males, 11 females, mean age: 23 years) at the Laboratory of Food Oral Processing, School of Food Science and Biotechnology, Zhejiang Gongshang University. All experiments were performed in compliance with the relevant laws, and an ethical approval was given for this study by the Research Ethics Committee of the Zhejiang Gongshang University (Approval Code: 2020041011). As a selection criterion, all participants were frequent milk drinkers, both of skimmed and full-fat milk. Assessors were asked to refrain from eating and drinking at least 2 h prior to sensory analysis, except for water. They received instructions regarding the evaluation procedure in both written and verbal formats, prior to sample evaluation. All assessors were comprehended about the triangle test and the rating system. A signed consent form was obtained from each subject before the experiment officially started. All experiments were carried out in a specially designated laboratory, where temperature and light intensity can be adjusted. The sensory test consists of two different parts:

### 2.6. Triangle Test and Intensity Scoring with Sensory Descriptors

In the first session, assessors were presented with two sets of milk samples simultaneously, each consisting of three samples of reconstituted milk or thickened fluids. Same mean droplet size, d_90_ = 0.3 µm, was maintained for each set of samples, but with varied fat content (either 2% or 7.5%).

In the second session, assessors were also presented with two sets of milk samples simultaneously, consisting of reconstituted milk and thickened fluids. Each set (with same fat content 7.5%, differed with droplet size,) consisted of three samples with the same fat content of 7.5 wt.%. Mean droplet size was set at either d_90_ = 0.3 µm or d_90_ = 3.5 µm, respectively. Two of the samples have the same fat droplet size, but the other one is different.

During sensory analysis, all assessors wore nose clips to prevent odor interference. Assessors were asked to sit still in a natural and comfortable manner, while they were guided to deposited 2 mL sample onto the top surface of the tongue. The assessors were asked to move the tongue tip against the palate in a parallel motion (to mimic a sliding movement), while ensuring no swallowing of the sample occurred. They were then asked to spit the sample into the wash basin after 3 s. Between each sample, bottled water was used for mouth cleansing. Assessors were allowed to repeatedly test the same sample, but trials should be no more than three times. After assessing all three samples, the assessor was then requested to determine which one was the odd sample. Assessors were also asked to discriminate samples using the attributes provided on the paper ballot and/or provide other sensory feelings they may experience. A six-point analytical rating scale with descriptions of degrees of each attribute was also provided, and subjects gave a mark for each sensory attribute on the scale. The ratings were then converted to a number between 0 (left) and 5 (right) (0 = not at all and 5 = very). Sensory attributes for assessment include smoothness, creaminess, and thickness. Smoothness was assumed to be proportional to the perceived friction force from the hard palate moving against the tongue surface. Creaminess was defined as the perception of ‘oiliness’ in the mouth and the degree of mouth coating. Thickness was defined as the sensed resistance to flow in the mouth.

### 2.7. Statistical Analyze

Statistical analysis was performed by SPSS software Version 25 (IBM, Chicago, IL, USA). The Shapiro–Wilk normality test was used to determine the normality; otherwise, a non-parametric test was used to analyze the differences between groups. Analysis of variance (ANOVA) and a nonparametric test (Mann–Whitney U Test) were used to determine the significance of difference between the friction coefficient and averaged sensory attributes.

## 3. Results and Discussion

### 3.1. Emulsion Characteristics

As the droplet size distribution might influence the texture and sensory perception, it is one of most important parameters that describe emulsion systems. The droplet size distribution data, for further analysis, was commonly presented by the form of its intensity and number as well as volume [35,36]. In this study, we presented the data with volume intensity for a clear observation. The droplet size distributions of reconstituted and thickened milk samples obtained after different homogenization procedures are shown in Figure 2, where volume density is plotted against the droplet size for selected milk samples (specific compositions of tested samples are shown in Table 1). As expected, the volume fraction of reconstituted milk samples (0.3, 0.7, 2, 3.5, 7.5 wt.% fat) homogenized using a shear homogenizer showed bimodal distribution, with its first peak occurring between 0.01–1 µm and its second peak between 1–10 µm. Thickened samples with fat content of 3.5 and 7.5 wt.% showed multimodal distribution with a third peak at 10–100 µm (Figure 2A,B). Figure 2C shows the fat droplet distribution of reconstituted and thickened milk samples after high-pressure homogenization. Typical monomodal distribution was observed for these samples, though a small tail was observable for samples containing 3.5% and 7.5% fat. The measured mean droplet size shows no statistical difference (*p* > 0.05) among these samples (see Figure 2C).

It is not yet fully certain what causes bimodality in Figure 2A,B. However, according to Laguna et al., the first peak in both the milk and thickened milk samples could correspond to free casein micelles, and the second one represents the fat globules [25]. Commercial dairy colloids, such as whole fat milk and milk cream, stabilized by milk globulin, are generally very stable, with almost no coalescence of the droplets in the presence of sufficient emulsifiers [37,38]. In the case of reconstituted milk, visual observations showed that no destabilization occurred over the timescales of the experimental tests. However, the addition of xanthan gum could lead to structural alteration to the colloidal system. It has been reported that, at a certain concentration, xanthan gum may promote flocculation of fat droplets and inhibit emulsification of fat by a depletion mechanism [39]. Thus, we intend to believe that the third peak in Figure 2B could be caused by the clusters of the fat droplets.

Figure 3 shows the flow curves of reconstituted milk and thickened samples, where the apparent viscosity of the milk sample is shown as a function of the applied shear rates. The rheology measurement was conducted over shear rates between 1 and 100 s^−1^, of similar order to the shear rate present in the tribological contact. Both types of systems exhibited a typical shear-thinning behavior, a decrease in apparent viscosity with increased shear rate, which is in good agreement with a previous study [40]. A milk sample, with a different fat mass homogenized using Ultra-Turrax (Shear), showed a weak shear-thinning effect and had a low viscosity (<0.01 Pa·s); an increase in fat mass fraction from 0.3 to 7.5% only had a marginal effect on viscosity increase (Figure 3A), which is in agreement with previous reports [25,28]. However, milk samples homogenized by HPH had significantly higher apparent viscosity compared with shear-homogenized samples. The former also had an obvious shear-thinning behavior, indicating that there are attractive interactions between the fat globules, probably caused by depletion forces induced by the casein micelles [41]. The flow curves of thickened milk are shown in Figure 3B. It is interesting to see that flow curves are very much overlapping; the addition of commercial thickener (xanthan) increased the viscosity of emulsion from 0.005~0.5 Pa∙s to 0.1~6 Pa∙s and controlled flow characteristics of the system. Compared with the unthickened samples in Figure 3A, the homogenization methods and fat content seem to have very limited effect on the rheology behavior of thickened samples. Table 2 shows droplet size and apparent viscosity at 50 s^−1^ for emulsion systems with and without thickening. One can see that the apparent viscosity only showed a slight increase (but with no statistical significance, *p* > 0.05) for a nearly two-magnitude reduction of fat droplet size.

### 3.2. Tribology: Lubrication Behavior of Milk and Thickened Milk Samples

Figure 4 shows the friction curves for all samples, in which the friction coefficient is plotted as a function of the sliding speed. We can see that two lubrication regimes, the boundary regime and the mixed regime, are clearly identifiable in most cases. The boundary regime at low sliding speeds has a friction coefficient nearly independent of the speed. At this regime, the load is supported by the asperity contact due to the absence of hydrodynamic pressure. At higher sliding speeds, the two surfaces start to separate from each other, and the lubrication effect starts to kick in, where a reduced friction coefficient becomes obvious for all sample systems.

#### 3.2.1. Effect of Oil/Fat Mass Fraction on Lubrication Behavior

Figure 4A shows the Stribeck analysis for the milk samples. It is interesting to note that skimmed milk has the highest friction coefficient. It is also interesting to note that the presence of a small amount of fat droplets (i.e., 0.3%) can lead to a significant reduction of the friction coefficient at the boundary regime. This seems to suggest that a thin layer of emulsion film is sufficient to create a significant lubricating effect at low sliding speed [28]. However, it is also clear that the thin layer is not sufficient at this regime to create a floating pressure against the load for systems containing 3.5% fat and lower, probably due to the low viscosity of such systems [21]. For the highest fat content, it was found that the friction coefficient was significantly lower than that of all other systems, over the whole investigated range of sliding speed.

At higher sliding speeds, the friction coefficient started to decrease due to the capability of emulsion lubricant to separate two surfaces with a higher hydrodynamic pressure. Nonetheless, as discussed in Section 3.1, the changes in viscosity were only fractional. Therefore, we intend to believe that lubricant viscosity is not the main cause for the observed reduction of friction coefficient. The transition from boundary to mix regime at lower sliding speeds for high fat content sample is caused by the entrainment of fat droplets between the two surfaces, and the lubrication effect became evident.

Figure 4B shows the friction curves for thickened milk samples containing various amount of fat droplets, where thickened water was also shown as a reference. A pronounced lubricating effect was observed for such systems, with (1) a much smaller friction coefficient; and (2) a much earlier transition (at a lower sliding speed) from the boundary regime to the mixed regime. Most shocking, friction reduction was observed for the one containing 7.5 wt.% fat. Its friction curve remains very low and flat over the whole range of sliding speed (from 0.1 mm/s to 20 mm/s), convincingly indicating the formation of an effective lubricating film between the two moving surfaces. It is, therefore, reasonable to conclude that, compared with the reconstituted milk system, a thickened system with an increased viscosity leads to a higher hydrodynamic pressure, which may promote the entrainments of fat droplets into the lubricating interface, creating a significant reduction of the friction coefficient. The above results seem to be in line with those obtained by Chojnicka-Paszun and de Jongh, who suggested that the addition of xanthan gum might mask surface roughness and improve lubrication [42]. Moreover, the study conducted by Bongaerts confirms the importance of lubricant viscosity on friction behavior; high viscosity is no doubt the main reason enhancing the lubricating effect [19], and this conclusion is consistent with the friction curve in Figure 4A,B (the sample had various viscosity under the same fat content). However, the friction curves were various among different samples even when the viscosity of the tested system is similar (Figure 3B), suggesting that the tribological property of emulsion is not only determined by viscosity.

#### 3.2.2. Effect of Oil/Fat Droplet Size on Lubrication Behavior

In addition to fat content, the fat droplet size has also been investigated for its effect on lubrication behavior, and the results are shown in Figure 4C,D. Bear in mind that these systems have rather similar rheological behavior and comparable shear viscosity (see Figure 3). However, it is striking to see hugely different Stribeck curves for these systems. The friction coefficient varies almost one order of magnitude at all experimental speeds. Larger droplet size seems to be beneficial and more efficient in friction reduction. For systems with similar viscosity (see Table 2), samples with larger fat droplets had a lower friction coefficient than those with smaller droplets as well as a prolonged boundary regime. This seems to further suggest that larger fat droplets are easier to be squeezed and entrained into the lubricant gap, and, as such, coalescence and accompanying film formation might occur more easily [32]. Investigations by Laguna et al. indicated that fat droplets could coalesce within the tribological contact surfaces, reducing the traction coefficient until a sufficiently high shear is established to disrupt any boundary layers [25]. Their result seems to suggest that, under inferior sliding speeds (compared to 1–1000 mm/s), boundary lubrication mediated mechanisms may not exist.

The results shown above demonstrate that increasing the mass content of fat droplets, with both a droplet distribution between 0.01 µm and 10 µm and droplet clustering (10–100 µm), can improve the lubricating effect. In order to identify the optimal distribution range of fat droplets for an enhanced lubricating effect, a series of milk and thickened samples with matching droplet size distribution have been fabricated using HPH with droplet size d_90_ = 0.3 µm (droplet size distributions are given in Figure 2D). Friction curves for the above-mentioned samples are shown in Figure 5, with two distinguishable patterns for the two different types of systems. The sample systems with a thickener addition show a much lower friction coefficient than their counterpart samples. However, for the same type of samples, the results seem to suggest that further reduction of droplet size may have little effect on the lubrication behavior. This observation seems contradictory to the previous report that dairy products with higher fat content should exhibit a superior lubricating effect [10,28]. Data in Figure 5 show that both systems tested under a controlled surface load of 0.0392 N do not appear to demonstrate a significantly lower friction coefficient with an increasing mass fraction of fat in both the boundary and mixed regimes, with no significant statistical difference (*p* > 0.5). For the effect of the speed of surface movement, the friction coefficient only shows a slight decrease with the increase of sliding speed.

Based on the above results, one may conclude that the size of fat droplets is closely associated with this frictional phenomenon. The effect of the presence of fat droplets appears to be rather surprising: the presence of a small amount of fat content, i.e., 0.3%, leads to a big friction reduction, yet a further increase in fat content of up to 7.5% gives no further benefit. Given this, we hypothesize that the presence of fat droplets even in a limited number leads to the entrainment between moving surfaces and may help to promote lubricating effect. We further speculate that once enough droplets for entrainment exist between the two surfaces, a further increase in the number of fat droplets gives no further benefit in surface lubrication. This could be the reason that there was no further friction coefficient reduction for a system with the fat content increased from 0.3% to 7.5%. A similar observation was obtained by previous studies [25,43]. However, these authors believed that the identical tribology curve was the result of using PDMS versus PDMS as contact material. As the current study also used PDMS as medium, and the friction curves were clearly distinguished for skimmed milk and milk containing 0.3% fat, we intend to believe that substrate material is not the dominating cause, while droplet size is.

#### 3.2.3. Relating Lubrication Behavior of Model Emulsion with Oil Droplet Size

The lubricating mechanism of milk fat droplet size was also conducted on vegetable oil (olive). Prior to lubrication study, the flow behavior of CSS-stabilized emulsions has to be investigated; the results are shown in Figure 6, so we can see that the apparent shear viscosity against the shear rate exhibits a near-Newtonian flow behavior for all studied systems. A mild influence of oil content and droplet size on shear viscosity is observable. However, all emulsion systems have a relatively low viscosity, particular at the higher end of shear rates. These results seem to be in contradiction with previous studies, where almost all emulsion systems appear to have a significant increase in apparent viscosity with the increase in fat content or fat droplet size [27,44]. The reason could be due to the different shear rate range for observation. At a low shear rate when there is no disruption of internal structure, the shear viscosity will generally show a very high value. Once internal structure is disrupted at a high shear rate, viscosity increase will usually be diminished.

The lubrication behavior of the above emulsions has been carefully studied, and the results are given in Figure 7, where the friction coefficient is plotted against the sliding speed. As show in Figure 7A, friction curves show a high similarity for all emulsion systems of the same droplet size distribution, yet with a different oil content (varying from 0.3% to 10%) (*p* > 0.05). All systems show a monotonous decrease with the increase of speed; however, no clear transition from boundary to mixed regime or from mixed to hydrodynamic regime could be observed. Compared to the work of Upadhyay, the measured friction was correlated negatively with the oil content (1–30%) and the friction coefficient was significantly different (*p* < 0.05), despite four types of emulsifiers being used [27]. It should be noted that the span of oil content in the study by Upadhyay was much higher compared with our work. A high concentration of droplets may alter the way of droplet entrainment during lubrication and, therefore, causes a considerable change to lubrication behavior.

The friction curves of emulsion with the same oil content but different droplet size distribution is shown in Figure 7B. It is interesting to see that, despite all emulsion systems showing a gradual decrease in the friction coefficient with the increase in sliding speed, the droplet size appears to have a significant influence on the friction coefficient. The friction coefficient shows significant decrease, when increasing the droplet size over the whole range of the sliding speed, even when the volume fraction was the same. From the data in Figure 7B, one may find that emulsions with a droplet size distribution d_90_ value at 1.5 and 2.8 µm have the optimal lubricating effect. Once the droplet size is larger than 4.5 microns, inferior lubrication seems to become the case. The exact reason is not yet clear. However, one possible explanation could be that large droplets may have difficulty for entrainment, in particular when droplets are close in size to the thickness of the lubricating film. Another possibility could be the limited number of available droplets for entrainment (since the oil content remained the same for all emulsion systems). Nevertheless, our results again suggest that appropriate droplet entrainment is the key for emulsion lubrication.

To take the results a step further, we combined the droplet size distribution and friction results together for discussion (as shown in Figure 8). As described in Figure 7A, the lubricating effect of nano-sized emulsion seems independent of fat/oil content; in other words, adjusting the volume fraction of fat/oil droplets in the white area in Figure 8A should not interfere with the friction curve. When the fat content is kept constant, increasing the percentage of large oil droplets (show as the patterned area in Figure 8A,B) facilitated the lubricating effect. However, once the droplet size reached d_90_ = 2.8, further increasing the droplet size in the system (blue area in Figure 8C) seems to have little contribution to the lubricating effect (Figure 8C).

Based on the results shown above, a schematic representation of the ‘lubricating fat droplets’ range was built, referred to as the red box in Figure 8C. The lubricating effect of samples with fat droplet size, mainly distributed on the left and right sides of the red box, appears to be poorer compared to samples whose fat/oil droplets are distributed within the red box. The distribution of the speculated range of oil droplets, which can facilitate the lubricating effect, is in accordance with the work by Upadhyay and Chen, with normalized oil droplet distribution (show in the insert picture in Figure 8C) and an increase in the oil mass fraction from 1% to 30% demonstrating a significantly lowered friction coefficient, although a different emulsifier was used [27].

### 3.3. Sensory Analysis

Discrimination and rating test analysis were used to collect sensory data; the three most-used sensory attributes and paraphrases were selected and provided to consumers. For a triangle test with total 22 responses, the minimum number of correct responses required for significance at *p* < 0.001 and *p* < 0.01 is 15 and 12, respectively (ISO 4120:2021 Sensory analysis—Methodology—Triangle test). Table 3 shows that the number of panelists who were able to discriminate between milk or thickened milk samples with constant fat content or fat droplet size was statistically significant.

As it can be observed in Table 3, for the discrimination ability by the panelists of reconstituted milk with different fat content but with the same fat droplet size (d_90_ = 0.3 µm), the number of correct responses just meets the significant level. In addition, for the other three sets, panelists showed a higher discrimination capability. The selected samples were successfully differentiated in terms of thickness, creaminess, and smoothness (*p* < 0.05). Figure 9 shows the average sensory attribute intensity scores with a standard error for all samples.

A sensory discrimination of samples with the same droplet size distribution but different oil content is shown in Figure 9A,C. The mean intensity rating of ‘smoothness’ did not show significant difference. This seems to agree with the lubrication results, where the friction-overlapped curves were observed for these samples (Figure 5). However, the emulsion of high oil content gives a significantly higher perception of ‘thickness’ and ‘creaminess’, with statistical significance. It suggests that perceived ‘thickness’ and ‘creaminess’ are the dominating sensory attributes used by panelists for discrimination of these emulsion samples.

Figure 9B shows sensory results for milk samples with the same amount of oil volume fraction (7.5%) but rather different droplet size distribution (d_90_ = 0.3 µm and 3.5 µm, respectively). Significant differences in perceived intensity of ‘smoothness’ (*p* < 0.05) were reported by the panelists, despite no significant differences in perceiving ‘thickness’ and ‘smoothness’. As expected, milk samples presenting high ranking in the ‘smoothness’ attributes depicted the lowest friction coefficient, as it was assumed to be negatively related to the frictional force caused by contact between the two surfaces [45]. However, this trend was not seen for further-thickened samples. As seen in Figure 9D, panelists appear to be unable to differentiate between the smoothness of the two samples, despite the significant difference in their friction behavior.

A phenomenon worth noting is that compared to a sample with fat content of 2%, panelists rated ‘creaminess’ significantly higher for samples with fat content of 7.5% (Figure 9A,C); reduced fat droplet size also showed an enhanced ‘creaminess’ sensation (Figure 9B,D) for samples with fat content of 7.5%, although it was not significant. It should also be noted that ‘creaminess’ is a very complex, multimodal sensory attribute involving olfactory, gustatory, and tactile cues [12]. Kokini first developed a texture-dominating model which describes ‘creaminess’ as a function of ‘smoothness’ and ‘thickness’ [45]. However, Chen and Eaton later indicated that the perception of ‘creaminess’ is strongly influenced by olfactory sensation [46]. Therefore, rather than a single stimulation, the perceived ‘creaminess’ should be a combined result or an integrated result of multiple sensations. Nevertheless, except for the negative correlation with droplet size-related data, a sample with a higher ‘creaminess’ score also showed a higher apparent viscosity (although not significant), which indicates a positive correlation between the ‘creaminess’ score and the rheology data, as observed by Sonne [47].

The results from the sensory test suggest that the panelists were able to differentiate samples with the same droplet size distribution but with different levels of oil/fat content, even though the perceived ‘smoothness’ may not play a dominant role in the discrimination. For thickened samples, almost all panelists were able to identify the difference within two sets of samples, mostly likely based on the perceived ‘thickness’ and ‘creaminess’. Moreover, it was noticed that the xanthan addition could cause an undesirable ‘slimy’ feeling, which may interfere with the sensation of ‘smoothness’ attribute as reported by a few participants. This may explain why milk samples with a similar ‘smoothness’ rating have a very different in vitro lubrication behavior.

We should also bear in mind that saliva is a significant factor in the oral processing of food and plays an important role in texture perception. During oral processing, oil droplets may experience flocculation due to the participation of salivary proteins. Even though the oral destabilization of food emulsions is beyond the scope of the current study, one should be aware that the different environment between in vitro tribological tests and real oral sensory analysis could also be an important reason for some discrepancy observed between the two sets of results.

## 4. Conclusions

Droplet size and oil/fat content are the two most important controlling factors for the design and production of food emulsion, playing an equally important role as surfactants, particular in dairy and other o/w emulsion products. How such factors affect the lubrication behavior and then oral sensation is a core question for oral lubrication research of food emulsions. For this reason, this study has carefully reconstituted sets of milk samples with carefully controlled oil/fat content and droplet size distribution as well as shear viscosity; their lubrication behavior was carefully analyzed. Sensory analysis has also been conducted, in trying to reveal the sensory meaning of the lubrication data. Obtained Stribeck curves gave a clear discrimination for emulsions that vary in oil/fat droplet size (but with constant fat content) and for emulsions that vary in oil content (but with the same droplet size). Besides, results from this study demonstrated that droplet size played an important role in the flow and friction behavior of o/w emulsion systems. An increase in the volume fraction of oil/fat droplets within a certain range exhibits a significant effect on the lubricating effect. Results from sensory analysis showed a majority of panelists were able to discriminate milk samples with different lubrication behavior.

## Figures and Tables

**Figure 1 foods-10-03024-f001:**
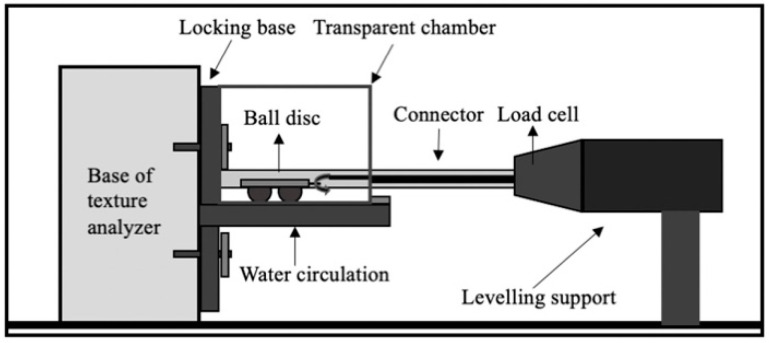
An illustration of the experiment setup ‘Soft Texture Analyzer Tribometer’.

**Figure 2 foods-10-03024-f002:**
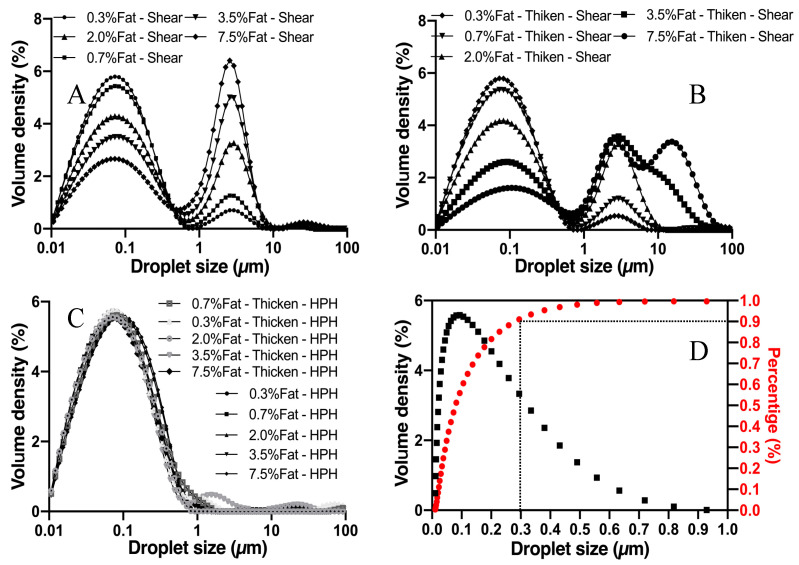
Droplet size distributions from the Malvern Mastersizer for reconstituted milk and thickened samples: (**A**) reconstituted milk homogenized under a shear homogenizer, (**B**) thickened reconstituted milk homogenized under a shear homogenizer, (**C**) reconstituted milk and thickened samples homogenized under HPH, and (**D**) droplet size accumulation curve.

**Figure 3 foods-10-03024-f003:**
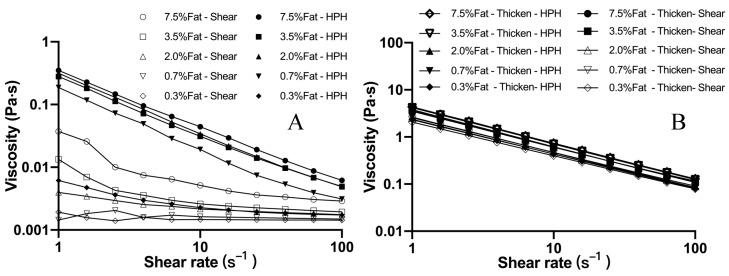
Flow curves for reconstituted milk (**A**) and thickened samples (**B**) with fat mass fractions of 0.3, 0.7, 2.0, 3.5, and 7.5 wt.%.

**Figure 4 foods-10-03024-f004:**
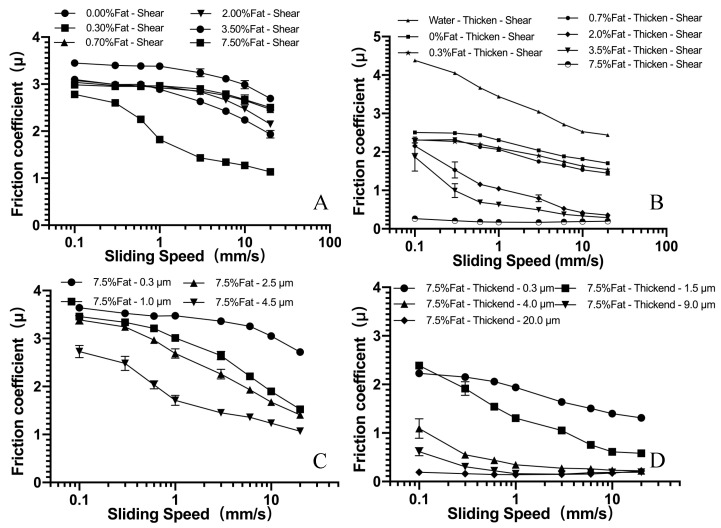
Stribeck curves for reconstituted milk and thickened samples, (**A**,**B**) with fat mass fractions between 0.3 and 7.5 wt.%, (**C**,**D**) with constant fat mass fraction of 7.5% and varied fat droplet size distribution. Lubrication tests were conducted at 25 °C and under a constant surface load of 0.0394 N.

**Figure 5 foods-10-03024-f005:**
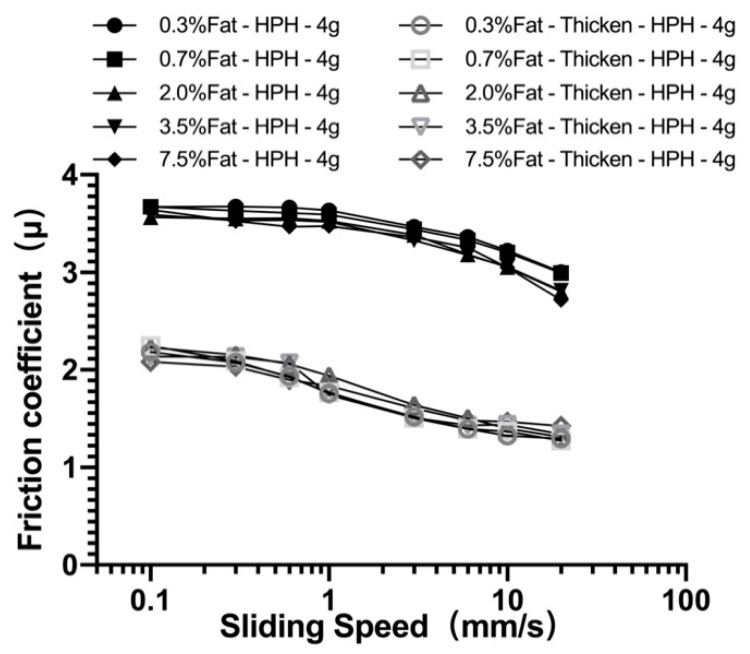
Stribeck curves for reconstituted milk and thickened samples with a constant fat droplet distribution and fat mass fractions between 0.3 and 7.5 wt.%. Lubrication tests were conducted at 25 °C and under surface load of 0.0394 N.

**Figure 6 foods-10-03024-f006:**
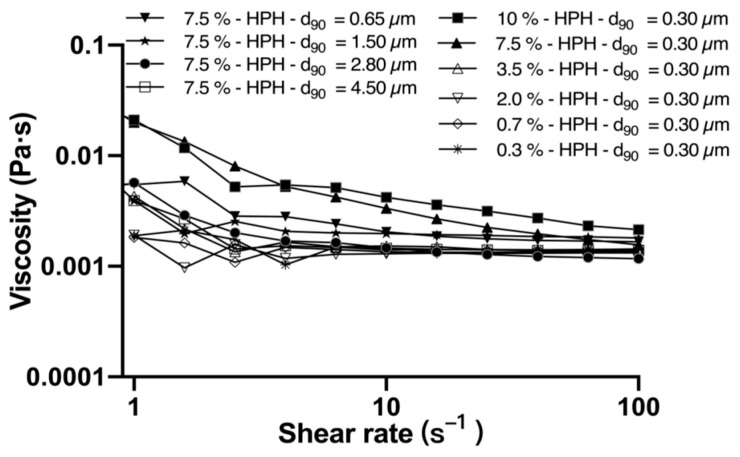
Flow *curves for CSS stabilized emulsions with oil* mass fractions of 0.3, 0.7, 2.0, 3.5, 7.5, and 10 wt.%.

**Figure 7 foods-10-03024-f007:**
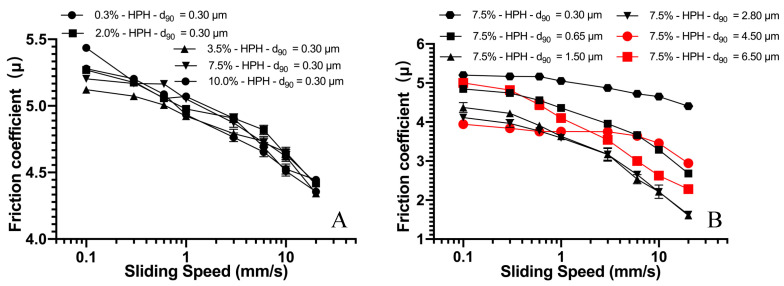
Stribeck curves for *CSS*-stabilized o/w emulsions with oil mass fractions between 0.3 and 10 wt.% with the same fat droplet distribution (**A**), with constant oil mass fractions 7.5% and varied oil droplet size distribution (**B**). Lubrication tests were conducted at 25 °C and under surface load of 0.0392 N.

**Figure 8 foods-10-03024-f008:**
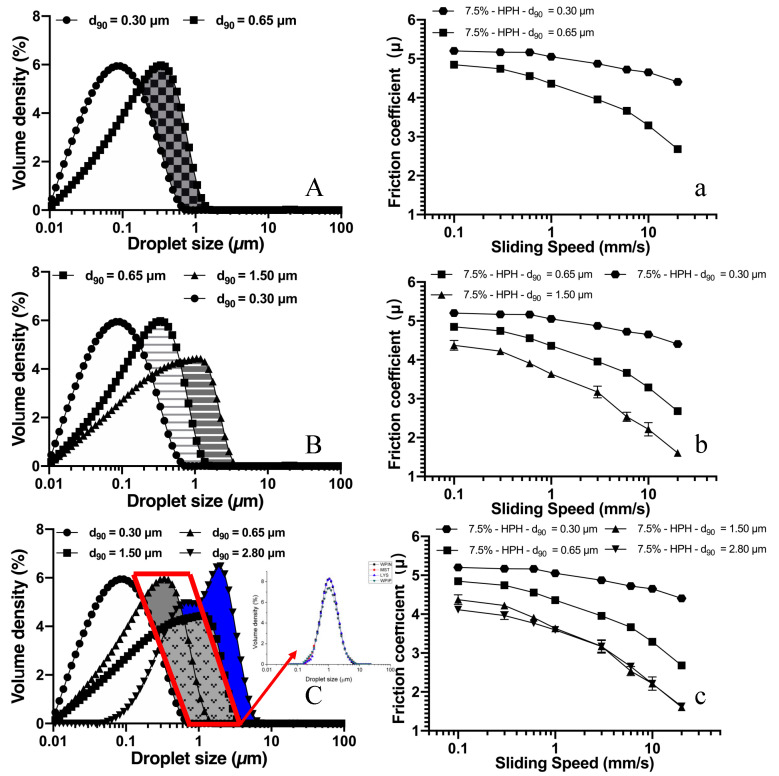
A schematic diagram showing the range of ‘lubricating fat droplets’ and its friction curves. Figures with capitalized letter shows the droplets distribution of emulsion sample with constant oil mass fraction but varied droplets size (**A**–**C**), figures with minuscules letter shows its friction curve (**a**–**c**).

**Figure 9 foods-10-03024-f009:**
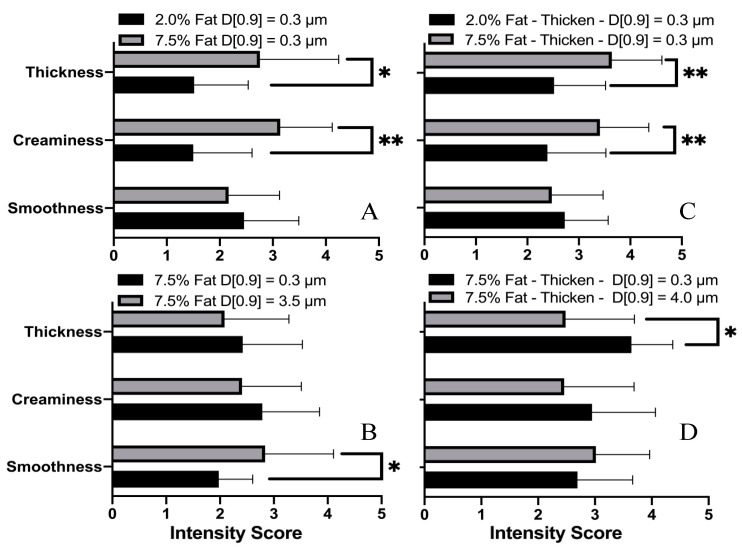
Ranking of selected sensory attributes obtained for samples with constant droplets size but with different fat content (**A**,**C**), or with same fat content but various fat droplet size (**B**,**D**). * Significance at 5%, ** Significance at 1%.

**Table 1 foods-10-03024-t001:** Composition of reconstituted milk and thickened samples.

Sample Name	Protein Concentration (wt.%)	Fat Concentration (wt.%)	Xanthan Concentration (wt.%)
0.3%Fat-Shear/HPH	3.74	0.30	None
0.7%Fat-Shear/HPH	3.57	0.70	
2.0%Fat-Shear/HPH	3.51	2.00	
3.5%Fat-Shear/HPH	3.44	3.50	
7.5%Fat-Shear/HPH	3.25	7.50	
0.3%Fat-T-Shear/HPH	3.74	0.30	0.86
0.7%Fat-T-Shear/HPH	3.57	0.70	
2.0%Fat-T-Shear/HPH	3.51	2.00	
3.5%Fat-T-Shear/HPH	3.44	3.50	
7.5%Fat-T-Shear/HPH	3.25	7.50	

**Table 2 foods-10-03024-t002:** Apparent viscosity (Pa∙s) measured at 50 s^−1^ of milk samples with constant fat content and various droplet size distribution.

	7.5 wt.% Milk		7.5 wt.% Thickened Milk	
	Fat Droplet Size (d_90_, µm)	Viscosity (50 s^−1^) (Pa∙s)	Fat Droplet Size (d_90_, µm)	Viscosity (50 s^−1^) (Pa∙s)
	4.617	0.00915	20.733	0.153
	2.370	0.00855	8.686	0.163
	0.990	0.00939	4.070	0.151
	0.299	0.0111	1.460	0.169
			0.301	0.213
Significance		None		None

**Table 3 foods-10-03024-t003:** Number of correct answers for sensory analysis using a discrimination test for milk and thickened milk samples.

Sample	Number of Correct Answers	Total Responses
2%/7.5% fat milk with same fat droplet size (d_90_ = 0.3 µm)	12 *	22
7.5% fat milk with different fat droplet size (d_90_ = 0.3/3.5 µm)	13 *	22
2%/7.5% fat thickened milk with same fat droplet size (d_90_ = 0.3 µm)	20 ***	22
7.5% fat thickened milk with different fat droplet size (d_90_ = 0.3/4 µm)	17 ***	22

* Significance at 5%, *** Significance at 0.1%.

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
