# Peer review of "Lubrication and Sensory Properties of Emulsion Systems and Effects of Droplet Size Distribution"

_foods, 2021, doi:10.3390/foods10123024_

Round 1
Reviewer 1 Report
You have done an extensive investigation. Congratulation for that.
- I think it could be very helpful to find in your paper an illustration of the used tribometer- please can you involve a simple illustration
- you present the range of the normal force. please could you give the range of the pressure in your friction tests
- I was surprised about the high friction coefficient! up to 4. please could you check if that is really right? Normally we find friction coefficients below 1. Please give an explanation for the high range if its real.
- you mention the region of the Stribeck-curve. Please could you describe in a sentence for one diagram were do you see the boundary- mixed- and liquid friction areas (ranges at the abscissa)
Reviewer 2 Report
The Authors presented a study focused on lubrication and sensory properties of emulsion systems and effects of particle size distribution. The idea of the examinations is interesting, but there is a lot of improvements and explanations needed before the final presentation. First, define what you mean by particles (see detailed comments). An additional figure which helps follow your ideas will be helpful. I will be grateful if you express more the examinations in the emulsion systems here:
3.3. Lubrication behavior of CSS stabilized o/w emulsions with Nano-scale oil droplet size and.
A major revision is needed. Detailed comments are listed below:
Abstract
- please cut the first sentence for a few. Now it is too much information in one place.
- define what you mean by "particles". Did you add solid particles into samples or did you study oil droplets - it must be clarified. Typically, emulsion systems use "droplet size distribution" (where there is no additional compound - particles)
- please add a description of your samples - what kind of emulsion did you study and what was the composition
Introduction
- l. 27 - the newest references are recommended
- use term"droplet size distribution" nor particle l. 37. As I good understood you mean here oil droplets.
- l. 46-47 what is the base of your hypothesis of "gold standard" - please add some references
- l. 78 - 79 these sentences are more suitable for the discussion section
- the last section of the introduction please add a description of your samples - what kind of emulsion did you study and what was the composition
Materials and methods
p.2.2 please clarify what you mean by samples here. Do you mean obtained emulsions? What was the dilution factor?
table with detailed samples description is necessary
Results and discussion
Why did you present data of droplet size distribution per volume? What about the droplet distribution by intensity and number? Did you use other techniques for the emulsion quality verification (you might see some examples here: Molecules 26(19):5856 or Molecules 25(11):2696
what do you mean by "bimodal" do you mean droplets fractions?
Please indicate in figure 3 "lubrication regimes, the boundary regime, and the mixed regime".
l. 353 please add some comments on what you mean by particles? - see previous comments
l. 357 you gave comments to fig. 1 from which it is difficult to find 300nm particles/droplets fraction (consider to add additional figure with no logarithmic scale)
l. 362 Against our general belief - rewrite this sentence. What do you mean by "belief"?
Enlarge fig 7C (inserted one)
l. 526 Droplet size and oil/fat content are the two most important controlling factors for the design and production of food emulsions - add-in which case O/W systems? What is more some other factor like stabilizers plays a crucial role. Please update your conclusions.
Round 2
Reviewer 2 Report
The Authors strongly improved manuscript quality. The responses to my questions are very satisfying. I m truly happy, that the Authors included graphical abstract which can be also inserted into the main text (not obligated) as a scheme 1 - it will clear your way of the performed research.
My recommendation is to accept.
One description should be changed in my opinion - not bimodal peaks, but peaks of the droplet fraction (in my opinion is more suitable here).